# Experimental Study of Cyclist' Sensitivity When They Are Overtaken by a Motor Vehicle: A Pilot Study in a Street without Cycle Lanes

Sebastian Seriani [1,*], Vicente Perez [2] , Vicente Aprigliano [1] and Taku Fujiyama [3]

1 Escuela de Ingeniería de Construcción y Transporte, Pontifica Universidad Católica de Valparaíso, Valparaíso 2362804, Chile
2 Facultad de Ingeniería y Ciencias Aplicadas, Universidad de los Andes, Santiago de Chile 7620001, Chile
3 Faculty of Civil, Environmental and Geomatic Engineering, University College London, Gower St., London WC1E 6BT, UK
* Correspondence: sebastian.seriani@pucv.cl

**Abstract:** The objective of this work is to study the sensitivity of cyclists who are overtaken by a motor vehicle in a street route without cycling lanes. To achieve this, an experimental method is proposed in which 17 cyclists were recruited and classified into two groups: non-experienced users and experienced ones. In each run, the motor vehicle generated a stimulus presented on the route, considering different speeds and distances in the overtaken maneuver of the motor vehicle. The sensitivity was reported by audiovisual records to capture the stimuli to which they were subjected. The results obtained show that an experienced cyclist reacts 1.27 times faster than a non-experienced one. This advantage gives experienced users greater safety and a level of confidence on the road, since being able to go faster, reduces their relative speed difference with motor vehicles and gives such cyclists a greater feeling of comfort during their trip. As future research, it is proposed to carry out studies with different types of cyclists and group size to be able to compare the perceived sensitivities of making the trip individually versus one made collectively for different types of road infrastructure such as dedicated cycling pathways.

**Keywords:** cyclist; safety; sensitivity; overtaken; motor vehicle; behavior

## 1. Introduction

As for other Latin American cities, Santiago de Chile has had a notoriously demographic growth in the last two decades. Consequently, new urban infrastructure megaprojects have been implemented to expand the city. However, these megaprojects are based favoring the use of cars by means of highways that have saturated in recent times. Nowadays, it is not uncommon to see that the population tends to use the bicycle as a preferred means of transport. This choice is related to the high travel times resulting from increased vehicular congestion that is observed especially at peak times, the high cost of public transport travel, or simply for a radical change in lifestyle towards sustainability and healthier life. This is illustrated by the study carried out by Jakovcevic, et al. [1], who studied to what extent the Public Bicycle Transport System (STPB) of the city of Buenos Aires had positive consequences for users when making their trips.

In 2020, Chilean the Ministry of Transport and Telecommunications (MTT) [2] conducted a study in which 1540 cyclists over the age of 18 were surveyed at 12 different points in the capital city (Santiago). The results indicated that 56% of those surveyed use the bicycle as a means of transportation daily and that 79.5% of the total trips made were intercommunal (i.e., between two or more boroughs). From the survey, 56.9% indicated that their motivation to use the bicycle came from the existence of road infrastructure that allowed them to use this mode of transport, 54% indicated that it was more comfortable for

them to move around the city by bike and 51.8% prefer the bicycle because it results in less travel time to reach their destination.

Comparing with a previous study by the MTT for the fifth World Cycling Index and published in 2019 by Eco-Counter [3] (a company specialized in counting bicycles and pedestrians worldwide) carried out between 2017 and 2018 in different cycle paths in the country, the trend continues and Chile is classified as one of the countries that has increased the use of this mode of transport reaching an annual rate that exceeds 10%, which is much higher than that of countries such as Austria, Switzerland, Australia and France, who are known for having cycle paths with much higher standards than those that can be found in the city of Santiago.

The implementation of infrastructure for cyclists may affect their behavior [4–6]. Cyclists differ from motorized vehicles in physical dimensions, weight ratio and travel speed, as well as in their maneuverability and ease of mobilization within a given track. Although it is urgent to determine which are the variables that describe the use of bicycles on streets without cycle lanes. Also, what is the difference in behavior between different types of cyclists for a given route? For example, what would be the difference in behavior in cyclists who have experience riding on a street compared to non-experienced users?

Therefore, the general objective of the work reported here is to study, experimentally, the sensitivity of cyclists in a street with mixed traffic to determine the response of users to real cases on routes without dedicated bike lanes. The specific objectives are: (a) define the singular variables that allow a better understanding of cyclists' behavior in routes without cycle lanes; (b) carry out in situ measurements of cyclists on a street without bike lanes in the city of Santiago, to adjust the variables that describe the average cyclist's movement; (c) define configuration scenarios for cyclists in coexistence with motor vehicles on roads without a dedicated bike lane; (d) analyze the behavior of different types of cyclists traveling along a defined route to study their responses to different stimuli to which they will be exposed.

In this study the behavior is analyzed according to the sensitivity of each cyclist, which is defined as a stimulus presented on the route when a motor vehicle overtakes them. This stimulus is classified into four aspects: the safety of the circuit, the confidence of each cyclist, the perception of the distance at which the cyclist was overtaken, and the perception of the speed with which the cyclist was overtaken.

The structure of the paper is divided into 5 sections. In Section 2, different studies on cyclist behavior are reported. Next, in Section 3 the experimental method is described. In Section 4 the results are analyzed, followed by the conclusions in Section 5.

## 2. Cyclist' Behavior

Although the bicycle was created in 1817 by the German inventor Karl Drais, which served as an effective mode of transportation at the time, no cycle-inclusive planning or study was needed for its implementation. At that time, there were conditions for pedaling since all the infrastructure for bicycles already existed in the form of main roads, therefore, one of the main technical difficulties was the absence of good pavement to achieve a good Bicycle Level of Service or BLOS [7], defined as the relationship between three main variables: flow (bicycle dynamics, hindrance, modal interaction), infrastructure (sharing policy, enforcement, pavement, trip-end) and exogenous variables (climate, topography, sociodemographics).

BLOS has been affected by the invention and widespread use of the automobile and the change in the landscape, as mentioned by the Department for Infrastructure, Traffic, Transport and Public Space of the Netherlands (CROW), in their Design Manual for bicycle traffic [8]. While there were 67,000 cars in the Netherlands in 1930, there were also 2.5 million bicycles on the streets. This encouraged a change in road networks because it made the speed contrast between a vehicle and a cyclist noticeable. This gave rise to the physical separation of both modes of transportation (cars and bicycles), applying engineering that sought to prevent conflicts rather than integrating bicycle traffic.

Similarly, Šemrov et al. [9] used BLOS to generate cycle lanes based on the comfort perceived by cyclists. The authors studied the width of the cycle lane, the speed of cyclists, traffic volume, among other variables.

In terms of standards, a cycle lane should be at least 2.0 m-wide (bidirectional flow) or 1.8 m-wide (unidirectional flow). These recommendations are established in Chilean Manuals, which have been recently launched [10]. These standards consider, on average, the width of the handlebar of a two-wheeled cycle of approximately 75 cm using an MTB-type bicycle (there are smaller variants such as route handlebars that reach a width of 42 cm). In addition, it considers that, when starting any movement on the cycle lane, there is a swing that oscillates approximately 25 cm and establishes that, when speed increases, a balance is reached, which, in turn, decreases the horizontal oscillation produced when starting the movement. This model of cyclists' behavior is accurate in the absence of additional stimuli during a journey, i.e., when cyclists are not overtaken by motorized vehicles or other cyclists. It can be said that a cyclist "attitude" is an internal process that mediates between an object (e.g., a motor vehicle) and the responses that a person (e.g., a cyclist) gives to that object. Although an attitude cannot be observed directly, it is believed that the positive or negative evaluation of an object carries with it the predisposition to respond in a certain way to it. For this reason, a relationship is established between the attitude towards an object and the reactions and behaviors that it arouses [11].

The behavior of cyclists depends mainly on the infrastructure capacity of the roads. Seriani et al. [12] analyzed flow levels in bicycle cycle lanes, to estimate if the capacity of the cycle lanes at saturated traffic lights. The authors concluded that there is an almost linear relationship between saturation flow and cycle lane width. In addition, it was observed that the capacity of the cycle lane increased if a new line of flow of cyclists was generated. In other words, cyclists will take over the physical space as long as its geometry allows them to transit.

If the most relevant variables such as the speed, acceleration and braking of a vehicle are analyzed, CROW [8] states that in physical terms, cyclists must overcome the forces of rolling resistance (determined by the type of tire), the porosity of the pavement, and the aerodynamic resistance, which will depend on the aerodynamic profile of the bicycle and the speed of the wind. As reference is made to resistance, it is necessary to make it clear that the mobility of the bicycle will be governed by the physical capacity of the user, that is, the cyclist is limited in terms of generating that energy. Therefore, a designed cycle-friendly road should cause the least amount of energy loss.

When there are roads without cycle lanes, motorized vehicles can change from one lane to another when overtaking a cyclist. Budhkar and Maurya [13] studied the lateral clearances maintained by the different types of vehicles as they move in a heterogeneous stream of traffic during overtaking. In other words, the variation of the free space obtained with the average speed of the interacting cars was studied and modeled. The authors concluded that pairs of similar vehicles maintained less lateral clearance than pairs of dissimilar vehicles. In addition, they added that when a vehicle interacts with two other cars on its sides (one on each side), both its safety and its lateral free space are reduced and compromised. This type of behavior has been also modeled by other authors [14], reaching similar results.

From a perception perspective, Muggenburg et al. [15] evaluate how people perceive different street designs for bicycle use through a questionnaire that presents three design options: conventional, flow and shared space. The first involves introducing traffic signs and on-street markings on a straight roadway. In the second, the travel modes are still separated by marking but in a roadway with slight curves. The third design is characterized by a space without elements that represent a physical separation of modes on-street. Through bivariate analysis and regression models applied to the survey results, the authors found that the shared space design for the cyclist is considered the safest, most fun and most attractive.

Under the argument that the existence of on-street barriers may be a relevant issue for cyclists that affects their perception of safety, Knight and Charlton [16] investigate through online questionnaires and on-road experiments the effect of protected and unprotected cycle lanes on cyclists' behavior and perception. The results of the study indicate that, generally, cyclists feel safer when there are barriers between bicycles and other modes of transport. In addition, people responded that, in a protected cycling lane context, they were willing to allow their children to cycle, because there is a lower accident risk.

Graystone et al. [17] investigate the gender gap in cycling, through a survey approach. They evaluate whether there are different perceptions from men and women towards streets with specific infrastructure for cycling and those without this infrastructure, in Toronto, Canada. The study shows that women tend to have more concerns about safety, however this does not affect, from a gender perspective, the fear of collision and injury. According to [17], the difference is highlighted regarding concerns related to verbal abuse and bullying in public spaces and how drivers interact with cyclists, mainly pointed out by women, independently of the available infrastructure.

Another research approach is presented by Silva et al. [18] by evaluating safety levels through a comparison of on-street and sidewalk bike lanes in Munich, Germany. This study applied inferential statistics of data related to interactions of cyclists in these two environments. Silva et al. [18] found that bike lanes on sidewalks generate more interaction between modes in comparison to on-street bike lanes. However, the latter shows a higher threat because of the potential interactions with vehicles at higher speeds, consequently generating lower safety for cyclists.

Skoczyński [19] seeks to present, using a planning perspective, a model that can support decision-makers to implement solutions for improving the safety of bicycle users on the road. That model is divided into three stages: diagnosis (I); measures planning (II); and implementation and monitoring (III). The first stage involves the analysis of cyclists' risks based on available accidents and collision data, followed by an assessment of the risk scale and characteristics of the risks. Furthermore, there is an evaluation of the current infrastructure and traffic organization from a quantitative and qualitative perspective. Then, stage II consists of a selection of prioritized areas for intervention with public consultation. These steps should lead to the pre-selection of possible solutions that could be presented in a second round of public consultation for their approval. The process ends with the implementation and evaluation of the effectiveness of the solutions.

Regarding overtaking maneuvers, the behavior of cyclists depends on different factors such as safety. According to Dozza et al. [20] the maneuver of a motorized vehicle overtaking a cyclist can be divided into four phases: approaching, steering away, passing, and returning. The phases are determined in relation to the strategy used to overtake the cyclist, which can be at a constant speed or an accelerative strategy in which the motorized vehicle stays behind the cyclist for a period of time before overtaking them. Experiments were done using an instrumented bicycle and Lidar data. In the case of cycle lanes in urban areas, the overtaking maneuver depends on the width of the road, the presence of other vehicles coming from an opposite direction and the presence of parking lots on the street [21]. Similarly, Pérez-Zuriaga et al. [22] studied the overtaking maneuver in rural environments, in which wider roads had a higher lateral clearance and overtaking speed, affecting the safety of cyclists. Another study done by Beck et al. [23] analyzed the space considered by motorized drivers in the city of Victoria, Australia, when overtaking a cyclist. The authors found that overtaking speed was affected by the presence of on-road bicycle lanes and parking lots on the street. In the case of intersections, Thorslund et al. [24] analyzed the speed, stops and trajectories of cyclists when motorized vehicles turn right or left at the intersection. The authors found that the type of motorized vehicle lane marking and availability of space adjacent to the motorized vehicle have an effect on a cyclist's behavior, speed choice and verbally expressed conscious strategies. Moreover, some authors [4] found, through a controlled laboratory experiment, the lateral displacement and speed

when interaction occurs in a maneuver. The authors also found that women cycle more slowly than men.

Through an investigation of 4 years of reported crash data in Beijing, Yan et al. [25] analyzed factors influencing motor vehicle and bicycle crashes. Results of this study showed that irregular maneuvers are related to age, gender, weather and traffic condition. Interestingly, the authors found that the speed limit has a significant influence on accidents, in other words, on high-speed limit roads, cyclists may have more difficulty acting in the face of an automobile threat at higher speed. This result also coincides with the recommendation delivered by Carvajal et al. [26], from a study developed on bicycle safety in Bogotá. The authors found that it is important to reduce the speed limit of motor vehicles in order to reduce accidents and their severity for bicycle users. Also, according to Chen et al. [27], the speed limit is an important factor that explains bicycle crash frequency, consequently affecting the safety of cyclists.

Furthermore, Isakkson-Hellman and Werneke [28], through data on insurance claims in Sweden from 2005 to 2012, found that 10.7% of crashes between bicycles and motor vehicles happen in the context that bicycles and cars are moving in the same or opposite direction in the road, and a significant number of these accidents are related to the speed of cars. Prati et al. [29] also point out that the speed limit is one of the factors that significantly influence the collisions between bicycles and motorized vehicles.

As a conclusion, it can be observed that the current literature is focused on the behavior of cyclists mainly when there are dedicated bicycle lanes or other type of infrastructure for cyclists. However, less research has been done to analyze the behavior of cyclists when there is a lack of dedicated cycle facilities such as in streets without cycle lanes, considering different speeds and distances in the overtaken maneuver of the motor vehicle which is the main objective of this study.

## 3. Experimental Method

In this section the experimental method is presented. Firstly, the variables to report the behavior of cyclists were defined, in which the sensitivity was registered. Secondly, the scenarios are described, considering the variation of the speed and distance in the overtaken maneuver of the motor vehicle. Thirdly, the data collection method is explained to process the experience of each cyclist during the experiment on the road.

### 3.1. Variables Observed

For this study, sensitivity is defined as the perception that a cyclist has at the time of completing their route considering the stimuli generated by the overtaking of a motor vehicle. Therefore, sensitivity is an emotional response which vary according to the ability and experience of the cyclist. This experience is a consequence of perceiving lack of safety during the journey (e.g., preventing them from continuing their journey along the route due to the high speed reached by motorized vehicles overtaking the cyclist), and therefore by having a perception of total confidence and fullness of maintaining the travel route without being affected by some external factor such as the overtaking maneuver of a vehicle. In addition, the sensitivity is affected by the perception of the distance at which the cyclist was overtaken, and the perception of the speed with which the cyclist was overtaken.

The route is taken to mean the infrastructure through which the user can move on the road without having a segregated cycle lane. The latter is transited by mixed traffic motorized vehicles. This means that there is physical movement of automobiles, motorcycles, public transport, cargo trucks and cyclists. In addition, this type of route (without cycle lanes) is characterized by the speed of movement that vehicles can reach (higher than that of a cyclist), which have a speed limit of 50 km/h in urban areas [30]. In this study the overtaking maneuver was done by a single motor vehicle.

The first variable analyzed is the sensitivity that cyclists experience when making their journey along a route without a cycle lane. As this will vary according to the condition and experience of each user, this sensitivity will be related to the response that cyclists

have when facing an external stimulus (i.e., overtaken by a motorized vehicle). Therefore, this will be the dependent variable selected for this work. This response will be quantified using a weighting, in which cyclists will be asked through a survey about the perception of their safety and confidence. This will be categorized using a scale of 1 to 5 (see Section 3.3).

The second variable studied is the distance that a motorized vehicle maintains at the time of overtaking a cyclist who is traveling on the lane nearest to the kerb of a route without cycle lanes. This will be defined as the first independent variable in the study. In other words, it is essential to observe if there is a relationship between the separation distance when a motorized vehicle overtakes a cyclist and the sensitivity that the cyclist may experience while riding the bicycle. This variable will be quantified in a magnitude of length ($L$). To this, the percentage of the total cases that perceived a value of $d$ less than the real $d$ with respect to the total responses is obtained as $P_d$.

It is important to bear in mind that the execution of overtaking a cyclist who is traveling on the street without segregation of cycle lanes is an action that compromises the integrity and safety of the latter during the maneuver. It should also be noticed that the driver of the motorized vehicle must respect and maintain an adequate distance when executing the maneuver. In this way, maintaining an appropriate distance when initiating the overtaking maneuver by the motor vehicle, will reduce the risk of causing an accident to the cyclist. According to the Chilean standards, the regulation is that all motorized vehicles must maintain a distance of at least 1.5 m (m) when starting, maintaining and finishing the overtaking maneuver [10]. Considering the free travel clearance for cyclists, plus the width of the lane they occupy and the overtaking distance required of 1.5 m as a safety standard, there will be events in which the overtaking motorized vehicle invades a portion of both lanes. In many cases, it may happen that the motor vehicle does not comply with current regulations and reduces the width of the travel lane carried by the cyclist. Therefore, this would be considered as a latent stimulus for the cyclist to travel, since it will cause the cycle user to either maintain their route or force them to move towards their right side. This, in turn, leads to exposing them to other stimuli belonging to the road infrastructure, such as the existence of drainage grilles at the nearside edge of the road which can generate an accident for cyclists (e.g., destabilize them).

The third variable of interest is the speed with which the motor vehicle overtakes the cyclist. As per distance, speed will be the second independent variable to be studied and will be quantified as the ratio between length and time ($L/T$). It will be observed if there is a direct relationship between speed (with which the motor vehicle overtakes the cyclist) and the cyclist's sensitivity. Also, it will be of interest to analyze the impact of the difference in travel speed between a motorized vehicle and a cyclist. In other words, it will be observed how the response of the cyclist manifests itself when being overtaken by a motorized vehicle that moves at a similar speed (e.g., 30 km/h). In addition, users' responses will be analyzed when they are overtaken at a speed higher than the speed limit (e.g., 60 km/h). Finally, the case will be studied for the normal speeds at which motorized vehicles move in the city of Santiago de Chile. It is important to note that the maximum speed allowed by the current traffic law in urban areas is 50 km/h. Similarly, the percentage of the total cases that perceived a value of $s$ less than the real $s$ with respect to the total responses is obtained as $P_s$.

### 3.2. Scenarios

To adequately specify the measurement of the variables identified in Section 3.1, the experiment will be carried out in an important street in the Chilean capital. The chosen street is transited daily by the inhabitants of the southern sector of the city for their trips to the different places of work, schools, and commerce, among others, because it connects several boroughs of the city of Santiago. The chosen place is Marathón Avenue (closer to the National Stadium), specifically the section between Los Plátanos and José Ananias streets.

This avenue meets the appropriate conditions to travel on a bidirectional paved road with a median separation. This avenue appropriately provides the stimuli (overtaken by a motorized vehicle) that can affect a cyclist's behavior on the street since it has two lanes for vehicles in a south-north direction and three lanes in a north-south direction, in the presence of mixed traffic (private vehicles and public transport buses, among others). For practical reasons and simplicity of measurements, the experimental study will be limited to only the right-hand side of the road (i.e., the south-north direction of Av. Marathón).

For the measurement, a circuit of 550 m length in Av. Marathon was designed, in which the speed $v$ of cyclists is registered. This begins at the intersection of Los Plátanos street and ends at the intersection of José Ananias street in the borough of Macul. This distance was estimated to represent a real situation that occurs daily in the transportation system of the capital when a cyclist travels on the right lane and is overtaken by a motorized vehicle. With this circuit, we wanted to ensure that the cyclist is not influenced by the experimental study. In this way, we seek to have a record that is as representative as possible of a real situation.

Therefore, the first 200 m of the circuit were considered so that the cyclist feels comfortable pedaling on the right lane of the street, reaching their desired speed. Thus, when arriving at the place of the yellow line demarcated on the pavement (see Figure 1), it is possible to perceive the real response that cyclists have before the stimuli to which they are exposed. The stimulus of being overtaken by a motorized vehicle was performed by the noise of the horn of the vehicle that moves next to the cyclist and that seeks to warn the cyclist that it is traveling next to them.

Another presented stimulus is the separation distance ($d$ in Figure 1 and Table 1) that is generated between the motorized vehicle and the cyclist when the overtaking maneuver starts. Different cases were studied. In the first case, $d$ was adjusted respecting the Chilean standards in which the motorized vehicle should keep a distance of 1.5 m when overtaking cyclists [10]. In the second case, the overtaking maneuver was carried out 1.1 m apart. With this distance, we seek to generate a direct comparison concerning the current minimum distance. And for the third case, the overtaking was carried out at 75 cm (cm) of separation. In this way, the cyclist was exposed to a real condition of overtaking, in which the driver of the motorized vehicle does not respect the minimum required standard.

Finally, the last stimulus was defined by exposing the cyclist to a motor vehicle that performed the overtaking maneuver at different speeds ($s$ in Figure 1 and Table 1). The first situation considered a speed $s = 30$ km/h. This speed limit has been implemented in different streets of the city of Santiago to allow and ensure safer trips for cyclists who travel through streets that do not have bike lanes. In a second situation, the overtaking maneuver was performed considering a limit speed $s = 50$ km/h. This value is the current speed limit for vehicles in urban areas. In the third and last situation, the overtaking action was carried out at 60 km/h. This value was chosen given that this has been reported as the speed limit in previous Chilean regulations of a few years ago [10].

To measure these stimuli a yellow line 10 m long was demarcated on the same street where cyclists had to travel (same direction as Av. Marathón, as shown in Figure 1). This demarcation was intended to guide the participants and prevent them from moving sideways while they pedaled. In other words, the aim was to prevent cyclists to be too close to the sidewalk, which is located on their right-hand side, as well as to prevent them from approaching the second lane of the street that is on their left-hand side. It was within that 10-m distance that the motorized vehicle did the overtaking maneuver considering each scenario of distance $d$ and speed $s$. To comply with the previously described scenarios in the overtaking maneuver, the motor vehicle had to make sure to pass over the demarked line in Figure 1, as this reference line had the function of guiding the driver to always keep the lateral distances $d$ constant with respect to the cyclist. In other words, by performing the test in this way, it was ensured that all cyclists were exposed to the same stimuli.

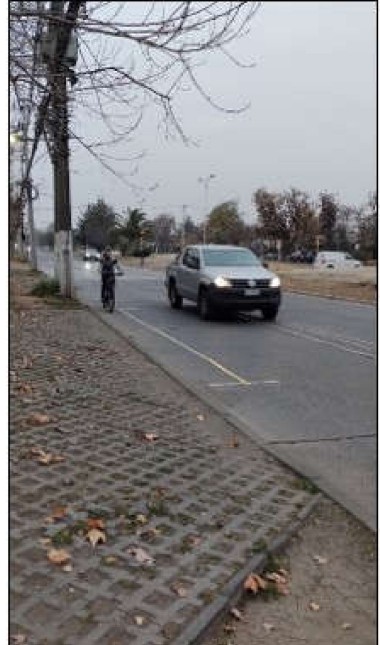
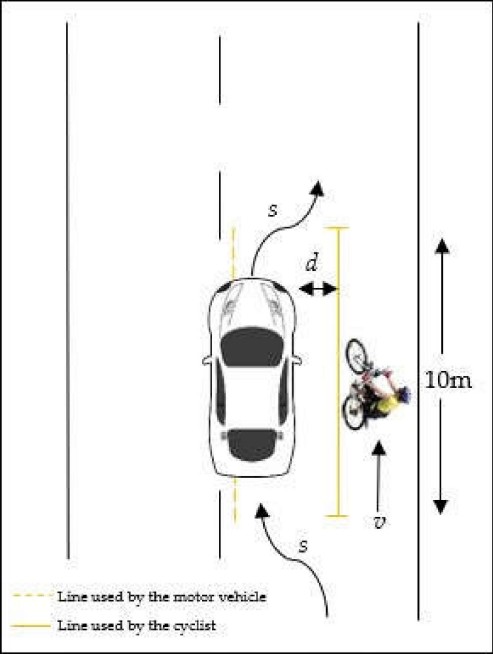

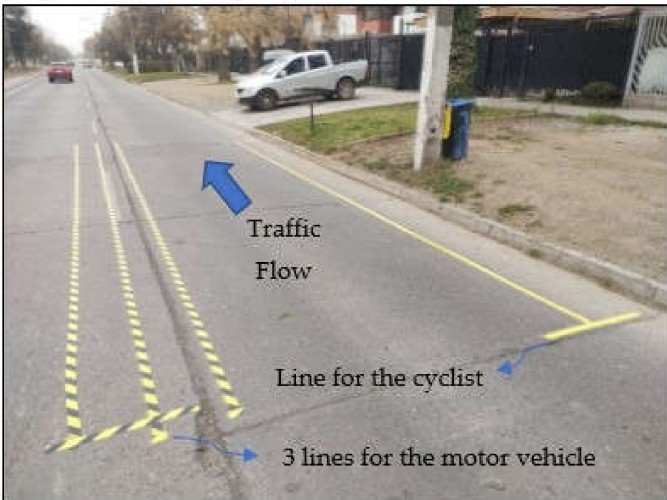

**Figure 1.** Av. Marathon in which observations were made when overtaking takes place in which *s* is the speed of the motor vehicle, *d* is the distances in the overtaken maneuver and *v* is the speed of the cyclist.

To capture and materialize the cyclist's sensitivity, a comparison was made between the movement of a non-experienced cyclist and an experienced cyclist, measured under two different modes of exposure to vehicular traffic (see Table 1). On the one hand, a non-experienced cyclist was classified as a user who is not used to and thus avoids riding on streets due to the dangers and demands that this entails. The main reasons for this behavior are related to the fear of being exposed to possible accidents and collisions with other vehicles, and the poor condition of the pavement. This last event is the product of the use and wear produced by heavier vehicles, which can cause instability in the cyclist and, therefore, cause a fall. This danger is heightened in the cases of cyclists with little experience making trips through the capital, as they tend not to organize or plan alternative routes to reach their destination.

In addition, it is assumed that the non-experienced cyclist has the tendency to travel at speeds less than or equal to the average of cyclists who circulate in bike lanes. This is based on the fact that, in general, their physical condition is not optimal, or that they are not normally accustomed to this type of physical demands (aerobic capacity and resistance).

Therefore, they prefer to travel using physically delimited and segregated sidewalks or bike paths, since those give greater priority to their safety and physical integrity. Concerning experienced cyclists, they were classified as users who have more than 2 years of experience riding the streets of the capital. In addition, they usually are physically fit enough to achieve a higher-than-average cyclist's road driving cadence (higher aerobic capacity and endurance mentioned when asking to participate in this study). This type of cyclist differs from the rest because they are cyclists whose movements are much more active than the average user. That is, they are capable of overtaking maneuvers on the streets, anticipating and previously calculating the time and distance of their movements.

A total of 17 cyclists (47% females and 53% males) voluntarily participated in the study. Seven of them were experienced cyclists (four of them made their journey with a road bike, one of them did it on a fixie-type bicycle and two of them did it with a mountain, MTB, bicycle). Likewise, it is observed that in non-experienced cyclists (a total of 10 cyclists) there is a clear tendency to use only MTB-type bicycles.

In the case of the motorized vehicle that participated in the experiment, a pick-up car model of the medium-size segment (Volkswagen Amarok model 2014) was used. This vehicle was selected because of its physical dimensions. In addition, it has the capacity and power necessary to perform acceleration and overtaking maneuvers on the street. In terms of space, the motorized vehicle occupied around 6 m long, 2 m wide, and 1.8 m height.

**Table 1.** Run scenarios to be studied on Av. Marathon.

| Run | Number of Experienced and Non-Experienced Cyclists | Distance of Overtaking $d$ (m) | Motorized Vehicle Speed $s$ (km/h) |
|---|---|---|---|
| 1 | 7 and 10 | 1.5 | 30 |
| 2 | 7 and 10 | 1.5 | 50 |
| 3 | 7 and 10 | 1.5 | 60 |
| 4 | 7 and 10 | 1.1 | 30 |
| 5 | 7 and 10 | 1.1 | 50 |
| 6 | 7 and 10 | 1.1 | 60 |
| 7 | 7 and 10 | 0.75 | 30 |
| 8 | 7 and 10 | 0.75 | 50 |
| 9 | 7 and 10 | 0.75 | 60 |

### 3.3. Data Collection Method

For the study of the scenarios defined in Section 3.2, a data collection method was chosen based on audio-visual records. To this, a camera adjusted to 1080p resolution was used, with a frequency of 50 FPS (frames per second). These settings were determined to store a lighter document, considering that the duration of the filming is approximately 1 min. Specifically, the camera was placed on the user's chest to more accurately represent the user's experience. That is, to capture in first person the view that the cyclist has when cycling down the street. In this way, it was possible to perceive and analyze most of the external stimuli that were generated during the journey.

For the registration of the speed reached in each test, the Strava application version 214.10 was used. It is available as a free download for iOS and ANDROID systems. Each cyclist was instructed to have the application installed on their mobile phone to complete the circuit. This application performs real-time monitoring using the GPS of the mobile phone. It provides the cyclist with details of the speeds reached (km/h) throughout the journey on a graph. It also shows the total distance traveled (km), the maximum speed obtained, among other variables in an easy way to analyze and compare results.

To optimize the data collection process, each cyclist was asked to restart the recording of the application for each lap of the circuit (see Figure 2). In this way, it was possible to have different records each time the cyclist completed a lap of the circuit. To measure the cyclist's sensitivity, they were asked to provide answers according to their driving experience on the circuit, marking only one option upon completion, to the following:

(a)   In terms of safety, cyclists needed to choose between these options: (1) The circuit is very unsafe and prevented me from completing the journey; (2) The circuit is unsafe, and despite achieving it I would not do it again; (3) indifferent; (4) The circuit is safe, I would do it again; and (5) The circuit is very safe, I had no complications doing it.

(b)   In relation to confidence, cyclists needed to choose between these options: (1) Very bad, it prevented me from finishing the journey; (2) Bad, despite achieving it, I would not do it again; (3) indifferent; (4) Good, I would do it again to acquire more confidence; and (5) Very good, it is a route that generates the confidence to be able to cycle.

With the results obtained from the experience of each cyclist, the options were weighed in order to quantify their travel experience by traveling on streets without a cycle lane.

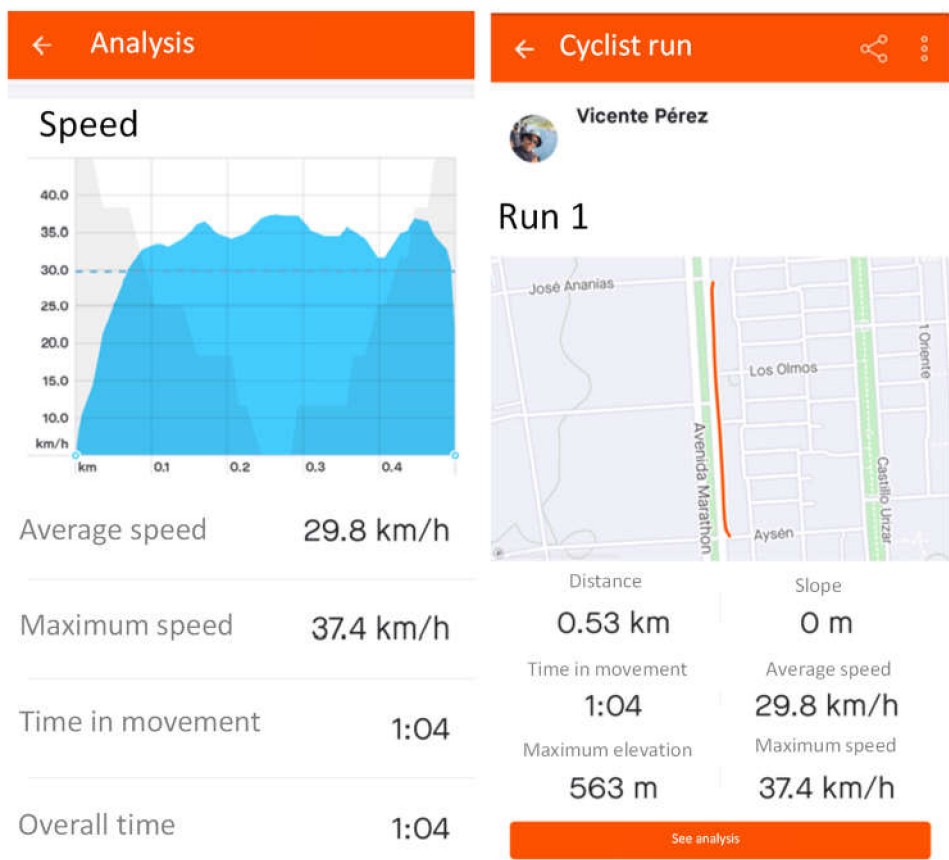

**Figure 2.** Screenshot of the Strava app used for the real-time monitoring of each cyclist.

## 4. Results

In this section the results are presented, which are classified according to the cyclist characteristics and the sensitivity reported by each volunteer who participated in the experiments.

### 4.1. Cyclist Characteristics

The experimental observations were carried out in Av. Marathón, on Saturday, August 14 and Sunday, 15 August 2021, during the COVID-19 Pandemic, in which 17 cyclists participated. From the total of cyclists, 8 of them were women and 9 of them were men. Each cyclist was classified according to their physical conditions, experience and performance on the route.

A group of 10 cyclists were classified as non-experienced, which represented 58.8% of the total. It is important to note that a greater participation of the female gender was classified as non-experienced cyclists (60%). In addition, it was observed that this group has an average age of around 30 years and their average height and weight is 1.69 m and

72.5 kg, respectively. It is also relevant that in this group (non-experienced cyclists) there is a greater preference for the use of MTB-type bicycles.

On the other hand, the group of 7 experienced cyclists, represented 41% of the total. At this point, a greater participation of the male gender (5 out of 7) stands out. Likewise, it was observed that this group of cyclists has an average age range of 28 years. In addition, the average weight and height of the group is around 70.1 kg and 1.70 m, respectively. In this group a greater tendency is observed for the use of road bicycles or similar.

These differences in the preference of different types of bicycles can be attributed to and associated with the choice of road infrastructure they choose when starting their trips on a normal journey to work or study. To this, a general survey was conducted with each volunteer to get a more detail insight into their choices. In the case of non-experienced cyclists, despite the fact that the majority (4 cyclists) preferred to use routes with cycle lanes, this is not a deciding factor for them. This is because they are willing to use other types of infrastructures such as the street or the sidewalk. In other words, 70% of cyclists who are classified as non-experienced avoid using the street as a route to make their trips and prefer other alternatives to complete their journey. The case of experienced cyclists is different, who clearly showed the tendency to use the street when making a cycling trip, as 6 volunteers expressed this preference and only one stated a preference for cycle lanes.

Regarding the reason why users use this mode of transport in the city, 69 % of the cyclists indicated that they cycle recreationally. Furthermore, when asked if they used the bicycle as a sport activity, 62.5% of them indicated this preference. They were also asked if they actually used the bicycle as their main means of transport, in which case 50% claimed to use this mode for work or study, and only one person indicated that it was their source of income (i.e., a delivery worker).

Another important aspect of these two categories of users was the time they had been using this mode of transport. In the case of non-experienced cyclists, 60% of these have been using it for less than 3 years, Furthermore, this type of user usually uses the bicycle only once a week.

On the other hand, in the case of experienced cyclists, it is observed that approximately 50% of them have been using this mode of transport for more than 3 years. It is worth noting that 43% of them had been using it for more than 7 years. Also of note is that in this group there is a clear tendency to cycle for 3 to 4 times a week. This clearly differentiates them from the behavior of non-experienced cyclists.

In addition, the average and maximum speeds reached by each of the cyclists in the completed laps (the cases explained in Section 3) are shown in Tables 2 and 3, respectively, for experienced and non-experienced cyclists. The figures were captured by each cyclist taking a screenshot of their smartphones' Strava displays.

**Table 2.** Average speed (*v*) of cyclists (km/h) recorded by Strava app for experienced (E) and non-experienced (N-E) cyclists by each run.

| Cyclist N° | Type | Run 1 | Run 2 | Run 3 | Run 4 | Run 5 | Run 6 | Run 7 | Run 8 | Run 9 | $\bar{X}$ |
|---|---|---|---|---|---|---|---|---|---|---|---|
| 1 | N-E | 18.7 | 20.8 | 17.9 | 17.3 | 19.3 | 19.1 | 19.4 | 18.6 | 19.9 | 19.00 |
| 2 | N-E | 23.3 | 21.6 | 23.2 | 24.0 | 22.5 | 24.9 | 24.8 | 23.2 | 24.1 | 23.51 |
| 4 | N-E | 16.5 | 16.5 | 15.9 | 15.6 | 15.3 | 16.4 | 15.6 | 15.8 | 15.7 | 15.92 |
| 7 | N-E | 14.6 | 16.4 | 14.0 | 17.1 | 17.3 | 17.9 | 16.2 | 17.1 | 17.3 | 16.43 |
| 8 | N-E | 20.3 | 20.3 | 19.1 | 20.0 | 21.1 | 19.3 | 22.4 | 19.4 | 21.6 | 20.39 |
| 9 | N-E | 15.3 | 15.1 | 17.2 | 17.0 | 17.2 | 15.7 | 16.0 | 16.5 | 18.4 | 16.49 |
| 10 | N-E | 16.5 | 17.0 | 18.1 | 18.6 | 18.0 | 16.8 | 18.3 | 17.5 | 17.7 | 17.61 |
| 12 | N-E | 17.5 | 21.0 | 19.1 | 20.4 | 21.1 | 19.1 | 19.0 | 20.2 | 21.1 | 19.83 |
| 13 | N-E | 20.2 | 21.3 | 20.8 | 19.4 | 19.8 | 16.3 | 26.3 | 20.1 | 22.5 | 20.74 |
| 14 | N-E | 24.5 | 22.2 | 19.0 | 22.8 | 17.7 | 24.4 | 23.1 | 25.4 | 17.8 | 21.88 |
| 3 | E | 28.3 | 28.6 | 26.7 | 20.8 | 27.6 | 24.3 | 26.3 | 25.3 | 24.3 | 25.80 |

**Table 2.** *Cont.*

| Cyclist N° | Type | Run 1 | Run 2 | Run 3 | Run 4 | Run 5 | Run 6 | Run 7 | Run 8 | Run 9 | $\bar{X}$ |
|---|---|---|---|---|---|---|---|---|---|---|---|
| 5 | E | 23.6 | 26.5 | 28.1 | 31.6 | 25.7 | 24.3 | 25.5 | 28.3 | 25.6 | 26.58 |
| 6 | E | 24.5 | 27.8 | 26.3 | 26.5 | 25.7 | 27.6 | 29.3 | 29.7 | 30.9 | 27.59 |
| 11 | E | 23.8 | 21.8 | 21.7 | 21.4 | 20.0 | 21.6 | 22.2 | 21.1 | 21.6 | 21.69 |
| 15 | E | 22.6 | 20.5 | 20.6 | 22.5 | 19.1 | 26.7 | 18.9 | 22.0 | 28.8 | 22.41 |
| 16 | E | 24.9 | 26.2 | 28.2 | 23.6 | 24.6 | 26.1 | 25.7 | 25.8 | 27.3 | 25.82 |
| 17 | E | 22.1 | 24.2 | 20.7 | 23.3 | 18.5 | 22.6 | 20.9 | 20.5 | 22.8 | 21.73 |

From the tabulated data in Table 2, the general average speed with which a non-experienced cyclist travels down the street is 19.2 km/h with a standard deviation of 2.8 km/h. Therefore, it is estimated that the speed fluctuations with which these users move will vary between 16 km/h and 22 km/h. In the case of maximum speeds (Table 3), non-experienced cyclists reached, on average, a top speed of 30.4 km/h, with a standard deviation of 5.4. That is, there will be cases in which these cyclists will move at speeds close to 35.8 km/h through the streets.

**Table 3.** Maximum speed ($v_{max}$) of cyclists recorded by Strava app considering experienced (E) and non-experienced (N-E) cyclists by each run.

| Cyclist N° | Type | Run 1 | Run 2 | Run 3 | Run 4 | Run 5 | Run 6 | Run 7 | Run 8 | Run 9 | $\bar{X}$ |
|---|---|---|---|---|---|---|---|---|---|---|---|
| 1 | N-E | 31.7 | 35.3 | 42.8 | 31.0 | 31.7 | 33.1 | 34.2 | 33.8 | 34.2 | 34.20 |
| 2 | N-E | 36.4 | 30.6 | 38.9 | 38.1 | 39.6 | 42.8 | 36.0 | 33.8 | 44.6 | 37.87 |
| 4 | N-E | 22.7 | 23.0 | 23.4 | 22.3 | 25.2 | 25.2 | 22.0 | 23.4 | 21.6 | 23.20 |
| 7 | N-E | 27.0 | 24.8 | 27.0 | 26.3 | 26.3 | 27.7 | 29.9 | 24.5 | 24.5 | 26.44 |
| 8 | N-E | 30.2 | 27.7 | 25.9 | 25.0 | 28.4 | 27.7 | 34.9 | 25.2 | 30.6 | 28.40 |
| 9 | N-E | 24.5 | 26.7 | 26.2 | 24.5 | 27.1 | 29.2 | 28.4 | 28.8 | 24.4 | 26.64 |
| 10 | N-E | 23.4 | 31.7 | 30.2 | 28.1 | 26.6 | 23.4 | 25.2 | 28.4 | 28.8 | 27.31 |
| 12 | N-E | 32.4 | 32.8 | 28.4 | 31.7 | 32.8 | 28.8 | 30.2 | 28.4 | 30.6 | 30.68 |
| 13 | N-E | 28.4 | 31.3 | 33.8 | 33.1 | 27.0 | 28.4 | 37.4 | 41.4 | 37.1 | 33.10 |
| 14 | N-E | 36.8 | 34.9 | 38.5 | 41.8 | 34.3 | 38.2 | 34.2 | 34.2 | 34.6 | 36.39 |
| 3 | E | 46.4 | 47.2 | 42.5 | 41.8 | 40.0 | 40.3 | 39.2 | 37.1 | 40.7 | 41.69 |
| 5 | E | 43.2 | 41.0 | 41.8 | 47.5 | 43.2 | 39.6 | 42.5 | 41.4 | 44.6 | 42.76 |
| 6 | E | 40.3 | 41.8 | 44.6 | 38.2 | 37.1 | 46.1 | 46.8 | 39.6 | 40.3 | 41.64 |
| 11 | E | 36.0 | 32.8 | 32.4 | 36.7 | 35.6 | 36.4 | 34.2 | 33.8 | 35.6 | 34.83 |
| 15 | E | 36.0 | 37.8 | 36.0 | 37.4 | 33.5 | 36.7 | 34.2 | 32.8 | 36.4 | 35.64 |
| 16 | E | 43.6 | 40.7 | 44.3 | 48.6 | 41.4 | 41.4 | 41.4 | 41.8 | 42.8 | 42.89 |
| 17 | E | 39.2 | 35.6 | 32.8 | 33.5 | 37.4 | 37.1 | 36.0 | 35.3 | 38.2 | 36.12 |

From the tabulated data in Table 2, the average speed for experienced cyclists is 24.5 km/h. In other words, this type of cyclist moves 1.27 times faster compared to the non-experienced users. In addition, a standard deviation of 3.1 km/h was recorded from the data, that is, the experienced cyclist will ride at speeds that will vary between approximately 21 km/h and 28 km/h. From Table 3, it can be seen that the average maximum speed reached by this type of cyclist is 39.4 km/h. If compared to the current maximum permitted speed for vehicles in urban areas, which is 50 km/h, this type of user travels 0.78 times slower on the street compared to motorized vehicles. In addition, it should be added that a standard deviation of 4.1 km/h was obtained from the data. This means that there will be cases in which these cyclists reached magnitudes of 43.5 km/h, thus reducing the speed difference with a motorized vehicle.

### 4.2. Cyclists' Sensitivity When an Overtaking Process Is Registered

Once the cyclists finished each run (or lap), they took a brief survey, that is, they had to answer the same survey a total of nine times. The purpose of this was to determine sensitivity to the stimulus of an overtaking motorized vehicle in a specific area of the circuit. This sensitivity was focused on four aspects as explained in Section 3: the safety of the circuit, the confidence of each cyclist, the perception of the distance at which the cyclist was overtaken, and the perception of the speed with which the cyclist was overtaken. To order and generate a clear detail of the above, the results are presented below.

In Table 4 the sensitivity of non-experienced by bicycle users is shown. From the data obtained in Table 4 it was observed that in the group of non-experienced cyclists, 44% of the total number of runs (or laps) completed indicated that the circuit was unsafe.

**Table 4.** Sensitivity of non-experienced cyclists to the safety of the circuit carried out on the street Av. Marathon.

| Run | The Circuit Is Very Unsafe | The Circuit Is Unsafe | Indifferent | The Circuit Is Safe | The Circuit Is Very Safe |
|---|---|---|---|---|---|
| 1 ($s$ = 30 km/h; $d$ = 1.5 m) | 0 | 2 | 3 | 4 | 1 |
| 2 ($s$ = 50 km/h; $d$ = 1.5 m) | 0 | 5 | 1 | 3 | 1 |
| 3 ($s$ = 60 km/h; $d$ = 1.5 m) | 0 | 4 | 2 | 3 | 1 |
| 4 ($s$ = 30 km/h; $d$ = 1.1 m) | 0 | 4 | 2 | 2 | 2 |
| 5 ($s$ = 50 km/h; $d$ = 1.1 m) | 0 | 5 | 1 | 3 | 1 |
| 6 ($s$ = 60 km/h; $d$ = 1.1 m) | 0 | 5 | 1 | 3 | 1 |
| 7 ($s$ = 30 km/h; $d$ = 0.75 m) | 2 | 4 | 1 | 3 | 0 |
| 8 ($s$ = 50 km/h; $d$ = 0.75 m) | 1 | 6 | 0 | 3 | 0 |
| 9 ($s$ = 60 km/h; $d$ = 0.75 m) | 1 | 5 | 1 | 3 | 0 |
| Total | 4 | 40 | 12 | 27 | 7 |
| Percentage | 4% | 44% | 13% | 30% | 8% |

In addition, from Table 4, it is interesting to identify that as the distance $d$ is reduced the perception of unsafety conditions increase. This is shown in the run N° 7, 8 and 9 in Table 4, in which zero participants considered that the circuit was "very safe" when the distance $d$ reached 0.75 m.

Moreover, from Table 4 it is observed the relationship between the speed $s$ and the perception of safety. Most of the cases in which s = 30 km/h reached a smaller number of responses considering the circuit "unsafe" or "very unsafe". Therefore, it seems that cyclists consider safer if the speed in which the motor vehicle overtake is lower than 50 km/h. If the speed $s$ is higher than 50 km/h more cyclists will consider that the route is unsafe.

On the other hand, in the case of experienced cyclists (see Table 5), the result of the sensitivity is different as 41% of the total number of runs (or laps) made indicated that they felt that the circuit was safe to travel by bicycle. This is directly related to what was described in Section 4.1, in which the profiles of each type of cyclist were analyzed, considering that experienced cyclists are more used to ride a bike on the street feeling safer even if there is no cycle lane infrastructure.

In comparison with the non-experienced users, it seems that the experienced cyclists did not make bigger changes in their perception of safety when the distance $d$ varied. In fact, Table 5 shows that the alternatives "very unsafe" and "very safe" presented zero responses in most cases. Similar situation is presented for the speed $s$, in which experienced users reach a speed $v$ which is in most cases similar to $s$ (see Section 4.1).

**Table 5.** Sensitivity of experienced cyclists to the safety of the circuit carried out on the street Av. Marathon.

| Run | The Circuit Is Very Unsafe | The Circuit Is Unsafe | Indifferent | The Circuit Is Safe | The Circuit Is Very Safe |
|---|---|---|---|---|---|
| 1 ($s$ = 30 km/h; $d$ = 1.5 m) | 0 | 1 | 3 | 2 | 1 |
| 2 ($s$ = 50 km/h; $d$ = 1.5 m) | 0 | 1 | 2 | 4 | 0 |
| 3 ($s$ = 60 km/h; $d$ = 1.5 m) | 0 | 2 | 1 | 3 | 1 |
| 4 ($s$ = 30 km/h; $d$ = 1.1 m) | 0 | 2 | 2 | 3 | 0 |
| 5 ($s$ = 50 km/h; $d$ = 1.1 m) | 0 | 3 | 1 | 3 | 0 |
| 6 ($s$ = 60 km/h; $d$ = 1.1 m) | 0 | 2 | 2 | 3 | 0 |
| 7 ($s$ = 30 km/h; $d$ = 0.75 m) | 0 | 3 | 2 | 2 | 0 |
| 8 ($s$ = 50 km/h; $d$ = 0.75 m) | 0 | 4 | 0 | 3 | 0 |
| 9 ($s$ = 60 km/h; $d$ = 0.75 m) | 0 | 1 | 3 | 3 | 0 |
| Total | 0 | 19 | 16 | 26 | 2 |
| Percentage | 0% | 30% | 25% | 41% | 3% |

In addition, another important aspect that was evaluated was the perception of the overtaking distance with which the cyclist was overtaken in the specific area of the circuit. From the above, the following results were recorded (see Tables 6 and 7). A direct relationship is observed between the nine cases studied and the sensitivity of the users. In other words, both types of cyclists (experienced and non-experienced) felt that, at the end of each circuit, the vehicle that was overtaking "pushed" or "forced" them towards the right-hand side of the road. In other words, it reduced the free space each cyclist used to ride their bike on the road. In fact, as can be seen in both Tables 6 and 7, in more than about 50% of the cases the cyclists felt that the motorized vehicle performed an overtaking maneuver less than 1 m away from them.

**Table 6.** Sensitivity of non-experienced cyclists to the distance of overtaking maneuver of a motorized vehicle.

| Run | Less Than 0.75 m | Between 0.75 m and 1 m | Between 1 m and 1.5 m | Between 1 m and 2 m | More Than 2 m | $P_d$ |
|---|---|---|---|---|---|---|
| 1 ($s$ = 30 km/h; $d$ = 1.5 m) | 0 | 2 | 3 | 4 | 1 | 50% |
| 2 ($s$ = 50 km/h; $d$ = 1.5 m) | 0 | 4 | 4 | 2 | 0 | 80% |
| 3 ($s$ = 60 km/h; $d$ = 1.5 m) | 0 | 1 | 6 | 3 | 0 | 70% |
| 4 ($s$ = 30 km/h; $d$ = 1.1 m) | 0 | 3 | 3 | 4 | 0 | 30% |
| 5 ($s$ = 50 km/h; $d$ = 1.1 m) | 3 | 4 | 3 | 0 | 0 | 70% |
| 6 ($s$ = 60 km/h; $d$ = 1.1 m) | 1 | 5 | 4 | 0 | 0 | 60% |
| 7 ($s$ = 30 km/h; $d$ = 0.75 m) | 2 | 5 | 3 | 0 | 0 | 70% |
| 8 ($s$ = 50 km/h; $d$ = 0.75 m) | 6 | 3 | 1 | 0 | 0 | 90% |
| 9 ($s$ = 60 km/h; $d$ = 0.75 m) | 7 | 3 | 0 | 0 | 0 | 100% |
| Total | 26 | 30 | 25 | 9 | 0 | |
| Percentage | 29% | 33% | 28% | 10% | 0% | |

$P_d$ = percentage of the total cases that perceived a value of $d$ less than the real $d$ with respect to the total responses

From Table 6 is observed the relationship between the separation distance $d$ when a motorized vehicle overtakes a cyclist and the sensitivity that the cyclist may experience while riding the bicycle. As the distance $d$ is reduced the perception it is shown a reduction of $P_d$, reaching the greatest value in the case of the run N° 9, in which 100% of cyclists perceived that the distance is less than $d$ = 0.75 m. The differences in $P_d$ could be affected not only due to the variation of $d$, but also because the speed $s$ of the motor vehicle is changed. However, in the case of experienced cyclists, Table 7 shows that the value of $P_d$

did not reflect the same variation than the case of non-experienced users. This could be caused because experienced cyclists are used to ride at a higher speed and therefore they may not be affected when changing the distance *d*.

**Table 7.** Sensitivity of experienced cyclists to the distance of overtaking maneuver of a motorized vehicle.

| Run | Less Than 0.75 m | Between 0.75 m and 1 m | Between 1 m and 1.5 m | Between 1 m and 2 m | More Than 2 m | $P_d$ |
|---|---|---|---|---|---|---|
| 1 ($s$ = 30 km/h; $d$ = 1.5 m) | 0 | 2 | 3 | 1 | 1 | 71% |
| 2 ($s$ = 50 km/h; $d$ = 1.5 m) | 0 | 3 | 3 | 1 | 0 | 85% |
| 3 ($s$ = 60 km/h; $d$ = 1.5 m) | 0 | 3 | 2 | 2 | 0 | 71% |
| 4 ($s$ = 30 km/h; $d$ = 1.1 m) | 3 | 2 | 2 | 0 | 0 | 71% |
| 5 ($s$ = 50 km/h; $d$ = 1.1 m) | 2 | 5 | 0 | 0 | 0 | 100% |
| 6 ($s$ = 60 km/h; $d$ = 1.1 m) | 1 | 3 | 3 | 0 | 0 | 57% |
| 7 ($s$ = 30 km/h; $d$ = 0.75 m) | 3 | 3 | 1 | 0 | 0 | 42% |
| 8 ($s$ = 50 km/h; $d$ = 0.75 m) | 4 | 3 | 0 | 0 | 0 | 57% |
| 9 ($s$ = 60 km/h; $d$ = 0.75 m) | 5 | 1 | 1 | 0 | 0 | 71% |
| Total | 18 | 25 | 15 | 4 | 1 | |
| Percentage | 29% | 40% | 24% | 6% | 2% | |

\*$P_d$ = percentage of the total cases that perceived a value of *d* less than the real *d* with respect to the total responses

As for the perception of overtaking distance, cyclists were asked to evaluate the perceived speed with which the motorized vehicle performed the overtaking maneuver for each run made. From the total number of runs made by the non-experienced cyclists (see Table 8), 38% of the cases cyclists felt that the vehicle overtook them at a speed of between 40 km/h and 50 km/h. This sensitivity of the cyclist is closely related to what was stated in Section 4.1, in which the speeds with which this type of user travels was analyzed. In other words, from the records obtained using the Strava app, these cyclists tended to travel at average speeds of 19.2 km/h. This implies that their sensitivity to this stimulus is felt more intensely. That is, the non-experienced cyclists perceived that the motorized vehicle overtakes them with a much greater speed difference compared to real speed of the motor vehicle.

**Table 8.** Sensitivity of non-experienced cyclists to the speed of overtaking maneuver of the motorized vehicle.

| Run | Less Than 30km/h | Between 30 km/h and 40 km/h | Between 40 km/h and 50 km/h | Between 50 km/h and 60 km/h | More Than 60 km/h | $P_s$ |
|---|---|---|---|---|---|---|
| 1 ($s$ = 30 km/h; $d$ = 1.5 m) | 2 | 1 | 6 | 1 | 0 | 20% |
| 2 ($s$ = 50 km/h; $d$ = 1.5 m) | 1 | 2 | 7 | 0 | 0 | 100% |
| 3 ($s$ = 60 km/h; $d$ = 1.5 m) | 1 | 0 | 7 | 2 | 0 | 100% |
| 4 ($s$ = 30 km/h; $d$ = 1.1 m) | 5 | 2 | 2 | 1 | 0 | 50% |
| 5 ($s$ = 50 km/h; $d$ = 1.1 m) | 1 | 5 | 3 | 1 | 0 | 90% |
| 6 ($s$ = 60 km/h; $d$ = 1.1 m) | 1 | 2 | 3 | 4 | 0 | 100% |
| 7 ($s$ = 30 km/h; $d$ = 0.75 m) | 1 | 5 | 3 | 1 | 0 | 10% |
| 8 ($s$ = 50 km/h; $d$ = 0.75 m) | 2 | 2 | 2 | 2 | 2 | 60% |
| 9 ($s$ = 60 km/h; $d$ = 0.75 m) | 0 | 4 | 1 | 3 | 2 | 80% |
| Total | 14 | 23 | 34 | 15 | 4 | |
| Percentage | 16% | 26% | 38% | 17% | 4% | |

$P_s$ = percentage of the total cases that perceived a value of *s* less than the real *s* with respect to the total responses

With respect to $P_s$ it is observed that there is not a clear tendency to determinate a relationship between the real speed $s$ and the perceived speed the cyclist has. Nevertheless, it could be mentioned from Table 8 that if the value of the distance $d$ is reduced then a greater number of cyclists perceived a higher speed compared to the real speed.

In the case of experienced cyclists, Table 9 shows the perception of cyclists was a little different from the non-experienced ones. In this case (experienced cyclists), 48% of them considered that the overtaking speed was between 30 km/h and 40 km/h. In this case (experienced users) most of the responses to the perceived speed were closer to the real speed of the motor vehicle. In this case (experienced cyclists), users are less affected by the speed of the motor vehicle compared to the non-experienced cyclists. In fact, in any run the speed was perceived as more than 60 km/h.

**Table 9.** Sensitivity of experienced cyclists to the speed of overtaking maneuver of the motorized vehicle.

| Run | Less than 30km/h | Between 30 km/h and 40 km/h | Between 40 km/h and 50 km/h | Between 50 km/h and 60 km/h | More than 60 km/h | $P_s$ |
|---|---|---|---|---|---|---|
| 1 ($s$ = 30 km/h; $d$ = 1.5 m) | 0 | 3 | 3 | 1 | 0 | 0% |
| 2 ($s$ = 50 km/h; $d$ = 1.5 m) | 1 | 5 | 0 | 1 | 0 | 85% |
| 3 ($s$ = 60 km/h; $d$ = 1.5 m) | 0 | 5 | 2 | 0 | 0 | 100% |
| 4 ($s$ = 30 km/h; $d$ = 1.1 m) | 2 | 5 | 0 | 0 | 0 | 28% |
| 5 ($s$ = 50 km/h; $d$ = 1.1 m) | 1 | 2 | 4 | 0 | 0 | 100% |
| 6 ($s$ = 60 km/h; $d$ = 1.1 m) | 0 | 2 | 4 | 1 | 0 | 100% |
| 7 ($s$ = 30 km/h; $d$ = 0.75 m) | 1 | 3 | 3 | 0 | 0 | 14% |
| 8 ($s$ = 50 km/h; $d$ = 0.75 m) | 0 | 3 | 4 | 0 | 0 | 100% |
| 9 ($s$ = 60 km/h; $d$ = 0.75 m) | 0 | 2 | 2 | 3 | 0 | 100% |
| Total | 5 | 30 | 22 | 6 | 0 | |
| Percentage | 8% | 48% | 35% | 10% | 0% | |

$P_s$ = percentage of the total cases that perceived a value of $s$ less than the real $s$ with respect to the total responses

With respect to the sensitivities of experienced cyclists (Table 9), from the total number of runs made, 48% of the cases cyclists experienced that the vehicle overtook them at speeds ranging from 30 km/h to less than 40 km/h. The data is directly related to the average speed with which this type of user travels (see Section 4.2). Since experienced cyclists have a higher average speed, their response is felt with less intensity. This is because the difference between the speeds of these cyclists and the vehicles that overtake them is smaller, and therefore there will be cases in which cyclists move at the same speed as the vehicles or even exceed the limit speed of 50 km/h.

The last sensitivity analysis to which the cyclists were subjected was to indicate their level of confidence at the time of completing each run of the circuit. Tables 10 and 11 show the responses of non-experienced and experienced cyclists, respectively.

From the results obtained from the survey, there are different answers for each type of user (see Tables 10 and 11). On the one hand, the group of non-experienced cyclists indicated that in 46% of the cases their level of confidence at the time of completing each run of the circuit was described as poor. This sensitivity responds to the years of experience they have as cyclists, the weekly frequency with which they use the cycle and the type of use they give it (as explained in Section 4.1).

**Table 10.** Sensitivity of non-experienced cyclists to the confidence on riding their bicycle.

| Run | Very Bad | Bad | Indifferent | Good | Very Good |
|---|---|---|---|---|---|
| 1 ($s = 30$ km/h; $d = 1.5$ m) | 0 | 5 | 2 | 2 | 1 |
| 2 ($s = 50$ km/h; $d = 1.5$ m) | 1 | 5 | 1 | 2 | 1 |
| 3 ($s = 60$ km/h; $d = 1.5$ m) | 0 | 4 | 4 | 1 | 1 |
| 4 ($s = 30$ km/h; $d = 1.1$ m) | 0 | 4 | 1 | 4 | 1 |
| 5 ($s = 50$ km/h; $d = 1.1$ m) | 0 | 5 | 2 | 2 | 1 |
| 6 ($s = 60$ km/h; $d = 1.1$ m) | 1 | 5 | 1 | 2 | 1 |
| 7 ($s = 30$ km/h; $d = 0.75$ m) | 1 | 4 | 2 | 3 | 0 |
| 8 ($s = 50$ km/h; $d = 0.75$ m) | 3 | 4 | 0 | 3 | 0 |
| 9 ($s = 60$ km/h; $d = 0.75$ m) | 2 | 5 | 0 | 3 | 0 |
| Total | 8 | 41 | 13 | 22 | 6 |
| Percentage | 9% | 46% | 14% | 24% | 7% |

**Table 11.** Sensitivity of experienced cyclists to the confidence on riding their bicycle.

| Run | Very Bad | Bad | Indifferent | Good | Very Good |
|---|---|---|---|---|---|
| 1 ($s = 30$ km/h; $d = 1.5$ m) | 0 | 0 | 1 | 4 | 2 |
| 2 ($s = 50$ km/h; $d = 1.5$ m) | 0 | 0 | 2 | 3 | 2 |
| 3 ($s = 60$ km/h; $d = 1.5$ m) | 0 | 1 | 2 | 2 | 2 |
| 4 ($s = 30$ km/h; $d = 1.1$ m) | 0 | 0 | 4 | 1 | 2 |
| 5 ($s = 50$ km/h; $d = 1.1$ m) | 0 | 2 | 2 | 2 | 1 |
| 6 ($s = 60$ km/h; $d = 1.1$ m) | 0 | 0 | 3 | 3 | 1 |
| 7 ($s = 30$ km/h; $d = 0.75$ m) | 0 | 1 | 3 | 3 | 0 |
| 8 ($s = 50$ km/h; $d = 0.75$ m) | 0 | 2 | 2 | 3 | 0 |
| 9 ($s = 60$ km/h; $d = 0.75$ m) | 0 | 0 | 4 | 3 | 0 |
| Total | 0 | 6 | 23 | 24 | 10 |
| Percentage | 0% | 10% | 37% | 38% | 16% |

On the other hand, in the category of experienced cyclists the feeling of confidence they experienced at the time of testing ranged from indifferent (37%) to good confidence (38%). As described in Section 4.1, these cyclists tend to make trips more frequently, preferably using the street. In addition, experienced cyclists consider the bicycle as their main mode of transportation, which is one of the main reasons why these cyclists are qualified as an experienced cyclist.

It is important to highlight that in the case of the speed $s$, it is not possible to identify a clear relationship. However, if the value of distance $d$ increased, non-experienced cyclists perceived a worst classification of confidence. This relationship for non-experienced users is different in the situation of experienced cyclists, which are less affected by the distance $d$ and speed $s$.

## 5. Conclusions

With this experimental study, it was possible to study the behavior of cyclists in a street to determine the sensitivity of users to real cases on a route without bike lanes. The experimental approach included 17 volunteers who were classified as experienced cyclists and non-experienced ones. The experiments allow us to change one variable "X" (e.g., speed of motorized vehicles) while the rest of them remain without variation. Therefore, it was possible to identify the effect of the variable "X" on the behavior of each cyclist. This experiment did not consider the analysis of the rest of the traffic as the main idea was to have a controlled environment to test "what if" scenarios.

Firstly, this study considered a more expanded definition of sensitivity in which cyclists were affected by the safety conditions of the route, and therefore it affected their confidence on riding a bike. But also, it was possible to identify a relationship between the perception of the distance at which the cyclist was overtaken and the real distance of overtaking, in which non-experimented cyclists were much more affected than experienced

ones. If the scenarios in which the distance was reduced, cyclists perceived that their distance was higher than the real one. However, it was not possible to obtain a clear relationship between the perception of the speed with which the cyclist was overtaken and the real speed of the motor vehicle, and therefore new experiments are needed to better understand the causes of these relationships.

Secondly, it is concluded that a description and categorization of the different types of bicycle users is needed to differentiate them in terms of the aspects involved for each cyclist when riding a bicycle. The most important aspects are the type of bicycle that they use, routes that they frequent, and their own safety assessment of the road infrastructure elements that they believe are necessary to achieve greater satisfaction and safety in each trip that they make.

Thirdly, it is relevant to mention that it was possible to recreate in a safe and controlled manner a case of a real "road coexistence conflict" that occurs daily on different streets and at different times in the city of Santiago de Chile. Specifically, it was possible to subject the different bicycle users to conditions that according to traffic regulations must always be respected, such as the speed limit in force in urban areas of 50 km/h and the minimum distance to overtake any bicycle that is 1.5 m. In addition, imposing critical conditions to these stimuli has been included to generate different responses in the users.

Fourthly, an interesting point is the responses of the volunteers who participated in the study from which it was possible to analyze their behavior on the street. Specifically, it was possible to perceive the differences in sensitivities that exist in the types of cyclists and how this response varies according to the conditions and external stimuli presented on a route without a cycle lane. In other words, given the same stimulus, the response between different types of cyclists can vary significantly, that is, what generates safety and confidence for a particular individual would not cause the same sensation in another.

With respect to the results obtained from the study, the following data are available:

- Non-experienced cyclists choose to use MTB-type bicycles since they provide different comfort conditions than a road bike, which is the bicycle most preferred by experienced ones since they favor speed.
- Less experienced users look for more alternative routes to get around, to avoid the street at all times. Conversely, experienced users do not "let go of the street", that is, at all times they choose to make use of this infrastructure for their trips.
- A difference in speed with which the different users move down the street was established. An experienced cyclist is 1.27 times faster than a non-experienced cyclist and on average manages to reach speeds of 25 km/h compared to an inexperienced one who travels on average at 19 km/h.
- In more than 50% of the overtaking tests carried out, both the non-experienced and experienced groups felt that a vehicle overtakes them at less than 1 m from them.
- Regarding the sensitivity of safety and confidence that can be achieved while traveling down the street, there is a greater sense of comfort if the trip is made at higher speeds (reducing the difference in speed with respect to a vehicle).

As a final conclusion, to feel a greater sense of safety on the street, cyclists (as some experienced users who participated in the study argued) must achieve a presence on the road and, as one cyclist commented at the end of one of the runs, "use more space on the street so that motorized vehicles begin to respect us". In other words, driving close to cyclists should be avoided, since more space is given for the vehicle to invade the "comfort zone" of cyclists and not respecting current traffic laws.

It should be considered that real-world experience by cyclists on the road may be different from the results of this controlled experiment. Furthermore, future studies could be conducted to compare random cyclists sampled in the field and the result of this experiment where cyclists knew they were being observed. In addition, future research requires considering group size and other types of cyclists on the road, such as delivery users, children, the elderly and disabled cyclists. These investigations will require new observation-based experiments in a larger number of types of streets which will be possible

in "normal conditions" considering that the current study was done in a COVID-19 scenario with limitations to minimize exposure to the virus.

**Author Contributions:** Conceptualization, T.F. and S.S.; methodology, S.S, V.P. and V.A.; software, V.P.; validation, S.S. and V.P.; formal analysis, V.P.; investigation, S.S., V.P., V.A. and T.F.; resources, S.S.; data curation, V.P.; writing—original draft preparation, S.S.; writing—review and editing, V.P., V.A. and T.F.; visualization, V.P.; supervision, S.S.; project administration, S.S. All authors have read and agreed to the published version of the manuscript.

**Funding:** This research received no external funding.

**Institutional Review Board Statement:** The study was conducted according to the guidelines of the Declaration of Helsinki, and approved by the Institutional Review Board (or Ethics Committee) of Universidad de los Andes (protocol code: CEC202089 and date of approval: 23 October 2020).

**Informed Consent Statement:** Informed consent was obtained from all subjects involved in the study.

**Data Availability Statement:** Not applicable.

**Acknowledgments:** The authors would like to thank the volunteers who participated in the experiments done in Av. Marathón, Santiago, Chile.

**Conflicts of Interest:** The authors declare no conflict of interest.

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
