# Peer review of "Experimental Study of Cyclist’ Sensitivity When They Are Overtaken by a Motor Vehicle: A Pilot Study in a Street without Cycle Lanes"

_sustainability, doi:10.3390/su142416784_

Round 1

Reviewer 1 Report

Recent events have highlighted the significance of the subject of this study. However, due to significant flaws in the way your research was conducted, I cannot support this paper.

1. I was wondering if you could look at some older research on collisions between bikes and automobiles and see if there's any way to relate it to what you've found.

2. Can you add just one paragraph to the third area (Experimental procedure) to summarize your experiment methodology, preferably with a diagram?

3. You limited the experiment to just one street in your observations. Is there any potential problem with the validity that your study area selection might bring?

4. You only have 17 cyclists on your squad. Is this all there is?

5. You merely performed a descriptive analysis. Is there enough evidence here to reach a conclusion?

6. Is the topic of this study truly the conflicts that arise between bikers and those who drive vehicles? How so?

7. The findings of previous research have already been confirmed to be comparable to those presented in the conclusion part of your paper. Then, what are some of the contributions that your paper makes?

8. What sorts of consequences does this have for public policy in your view?

Author Response

Dear Reviewer,

Please find attached the answer to your comments.

Best wishes,

Sebastian

Reviewer 2 Report

1. I would like to know why you did not use random cyclist passing your study area who are not aware that they are being studied and observed in order not to affect their actual behavior while using the road facility without bike lanes? Since you asked volunteer cyclists to be experimented and observed, this conscious awareness of being observed will be totally different if they are not aware of being observed. Can you expound why your approach of using volunteers and not random cyclists on the road?

2. Provide the statistical significance of your data especially for tables 2 & 3. Your number of samples may not be enough to provide a good statistical significance. Is it possible to add more samples?

3. In your recommendations, try to provide and discuss the limitations of your study and how this can be improved.

4. You did not provide discussion about the traffic mix or volume of traffic present in the area which might have also affected the cyclist behavior. Please discuss.

Author Response

(The authors gave the same response as above.)

Reviewer 3 Report

 Dear Authors,

The paper is very well structured. The content and theme of the article is consistent with the lines of the journal and the topic is of interest to the readers.

However, the paper has some inconsistent compering to the instructors for the authors, which should be corrected:

Overall the presentation is reasonably good, but it might still require some work:

·         The title of the manuscript are concise, specific and relevant. This is Ok.

·         The Abstract contain less than 200 words. This is Ok.

·         The Abstract contain all main obligatory elements (according to the instruction to the authors): Background of the research, Methods; Results; Conclusion. This is Ok.

·         List of Keywords is appropriate.

·         The paper contain all main obligatory chapters (Introduction; Materials and Methods; Results; Discussion). This is Ok.

·         Introduction chapter contain all mandatory elements such are: define purpose of the work, defining specific hypotheses which have being tested, current state if the research field, key publication from the filed cited. However, main conclusion are not presented. Should be added.

·         Material and Methods chapter in detail describe provided research with sufficient information for replication of provided research.

·         The Result chapter provide concise and precise description of the experiment results.

·         The Discussion chapter is integrated with Conclusion chapter and present results of the research. This is Ok.

·         The references are numbered in order of appearance in the text in the text in square bracket. This is Ok.

·         All equations are numbered in brackets and placed on the right margin of the text. This is Ok.

·         After every chapter name (title) should be some entrance sentence. It is not appropriate to have chapter title followed with chapter subtitle (e.g chapter title 3 and 3.1; 4.2 and 4.2.1; 4 and 4.1; etc. All of them do not have any description between chapter titles and subtitles).

·         In the paper is often used expression “in this study” or “for this experimental study”. However, instead of study authors should use expressions like “research” or “in this paper” etc.

·         In the paper is often used expression “we”. The paper must be prepared in third face singular.

·         Beside that the authors present very interesting research number of bicyclist used for testing is not significant (only 17 bicyclist in 9 scenarios/it is very small).

·         Authors should place explanation of every table/figure in front of them (not after them. Consider for correction.

·         It is not appropriate that two tables follow one another without any explanation between them (Table 4 and Table 5). Should be corrected.

·         Some tables are divided on two pages, with only column names on one page and rest of the table on anther page (Table 5; Table 7, Table 8). Should be corrected.

Author Response

(The authors gave the same response as above.)

Round 2

Reviewer 1 Report

Thanks for addressing my comments.

Author Response

Thanks for your contributions and valuable suggestions.

Reviewer 2 Report

Maybe provide a statement that real world experience by cyclist on the road may be different from the results of this controlled experiment. Another study could be conducted to compare random cyclist sampled in the field and the result of this experiment where cyclists knew that they are being observed.

Author Response

Thank you for the valuable suggestions. In addition, we added the present suggestion in the final part of the conclusion section. Which was the following statement: “It should be considered that real-world experience by cyclists on the road may be different from the results of this controlled experiment. Furthermore, future studies could be conducted to compare random cyclists sampled in the field and the result of this experiment where cyclists knew they were being observed.".